# DiffStack: A Differentiable *and* Modular Control Stack for Autonomous Vehicles

**Peter Karkus**[1], **Boris Ivanovic**[1], **Shie Mannor**[1,2], **Marco Pavone**[1,3]
[1]NVIDIA Research, [2]Technion, [3]Stanford University
{pkarkus,bivanovic,smannor,mpavone}@nvidia.com

**Abstract:** Autonomous vehicle (AV) stacks are typically built in a modular fashion, with explicit components performing detection, tracking, prediction, planning, control, etc. While modularity improves reusability, interpretability, and generalizability, it also suffers from compounding errors, information bottlenecks, and integration challenges. To overcome these challenges, a prominent approach is to convert the AV stack into an end-to-end neural network and train it with data. While such approaches have achieved impressive results, they typically lack interpretability and reusability, and they eschew principled analytical components, such as planning and control, in favor of deep neural networks. To enable the joint optimization of AV stacks while retaining modularity, we present DiffStack, a differentiable *and* modular stack for prediction, planning, and control. Crucially, our model-based planning and control algorithms leverage recent advancements in differentiable optimization to produce gradients, enabling optimization of upstream components, such as prediction, via backpropagation through planning and control. Our results on the nuScenes dataset indicate that end-to-end training with DiffStack yields substantial improvements in open-loop and closed-loop planning metrics by, e.g., learning to make fewer prediction errors that would affect planning. Beyond these immediate benefits, DiffStack opens up new opportunities for fully data-driven yet modular and interpretable AV architectures. Project website: https://sites.google.com/view/diffstack

**Keywords:** Differentiable Algorithms, Autonomous Driving, Planning, Control.

## 1 Introduction

Intelligent robotic systems, such as autonomous vehicles (AVs), are typically architected in a modular fashion and comprised of modules performing detection, tracking, prediction, planning, and control, among others [1, 2, 3, 4, 5, 6, 7, 8]. Modular architectures are generally desirable because of their verifiability, interpretability and generalization performance; however, they also suffer from compounding errors, information bottlenecks, and integration challenges.

A promising line of work tackling these issues focuses on making AV stacks more integrated (by relaxing inter-module interfaces) and data-driven (by optimizing modules jointly with respect to their downstream task). For example, in the context of AV perception, recent work has achieved substantial performance gains by jointly training tracking models with detection [9] and prediction models [10, 11]. To extend such a joint, data-driven approach to decision making, existing approaches replace hand-engineered components, e.g., planning and control algorithms, with deep neural networks [12, 13, 14]. As neural networks are differentiable, they can be optimized end-to-end for a final control objective; however, they offer weaker generalization, little to no interpretability or safety guarantees.

We introduce **DiffStack**, a differentiable AV stack with modules for prediction, planning, and control that combines the benefits of modular and data-driven architectures (Fig. 1). The prediction module in DiffStack is a learned neural network that predicts the future motion of agents; the planning and control modules are principled, hand-engineered algorithms that produce AV actions given the current world state and motion predictions. Importantly, our hand-engineered planning and control algorithms are *differentiable*, enabling the training of the upstream prediction module for a downstream control objective by backpropagating gradients *through* the algorithms. In doing so, DiffStack can jointly

6th Conference on Robot Learning (CoRL 2022), Auckland, New Zealand.

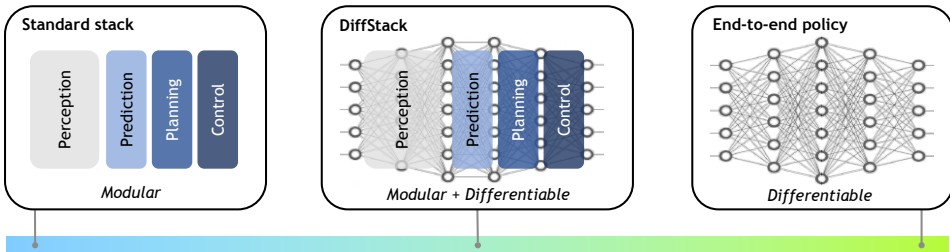

Figure 1: We desire AV architectures that are modular *and* differentiable, combining interpretability, reusability, and generalizability with data-driven end-to-end training. We take an important step towards this end with DiffStack, a differentiable AV stack for prediction, planning, and control.

optimize the entire stack for the final control objective, as in end-to-end neural networks, however it also maintains the interpretability and formal guarantees of standard modular AV architectures.

To make the planner and controller differentiable, we build upon the growing body of work on differentiable algorithm networks [15]. In particular, we leverage the differentiable Model Predictive Control (MPC) algorithm [16] for our controller. While differentiable algorithms show great promise, most prior works consider relatively simple control problems, e.g., pendulum and cartpole [16]. To the best of our knowledge, this is the first work to demonstrate that, through careful design choices and integration, differentiable algorithms can be composed into stacks and trained with real-world data for AV prediction, planning, and control.

We evaluate DiffStack in both open-loop and closed-loop simulation settings using the large-scale, real-world nuScenes dataset [17]. Our results show some immediate benefits of differentiable stacks: by training a prediction model with respect to the final control objective, DiffStack increases the effectiveness of predictions for decision making by up to 15% over a large number of diverse scenarios. DiffStack achieves this by, e.g., learning to make fewer prediction errors that would negatively affect planning. Further, DiffStack outperforms alternative handcrafted planning-aware prediction losses; compensates for artificially introduced integration errors; and tunes an interpretable cost function to make resulting plans more similar to human driving. Beyond these immediate benefits, our work demonstrates the feasibility and potential of differentiable AV decision making stacks, making an important step towards a new class of data-driven yet modular AV architectures.

## 2 Related work

**Modular AV architectures.** Many state-of-the-art methods for AV perception have achieved substantial gains by relaxing bottlenecks between learned neural network modules and training multiple modules together. Examples include works that more tightly integrate object detection and tracking [9], as well as prediction with object detection [11], classification [18], and tracking [19, 20, 10]. While successful, these works focus on AV modules where neural networks are already commonly employed and thus differentiable, i.e., perception and prediction. Looking downstream, there are many works that focus on more tightly integrating prediction within planning [21, 22, 23, 24, 25], however, these works only *use* predictions in planning rather than *improve* the prediction module with information from planning. Recently, works such as [26, 27, 28] present potential avenues for designing upstream modules in a planning-aware fashion. In particular, Ivanovic and Pavone [27] propose an evaluation metric for motion prediction that considers the impact of predictions on downstream planning. While this metric can be used to train prediction with information from planning, we will show in Section 4 that the resulting improvement is less than with DiffStack.

**Data-driven AV architectures.** Another line of work eschews modularity entirely, and learns a policy network directly from data [12, 29, 30]. The approach benefits from the removal of information bottlenecks and performance scaling with increasing dataset sizes. However, fully end-to-end approaches lack interpretability, important for debugging, verification, and safety guarantees; it is also unclear how well purely data-driven methods can generalize to unseen or rare-but-critical scenarios. Accordingly, recent works have focused on reintroducing structure and interpretability to learned policy networks. Most notably, the Neural Motion Planner [13] and follow-up methods [31, 32, 14] use object detection and agent behavior prediction as interpretable intermediate auxiliary objectives while training a joint backbone network from sensors directly to decision making. An important

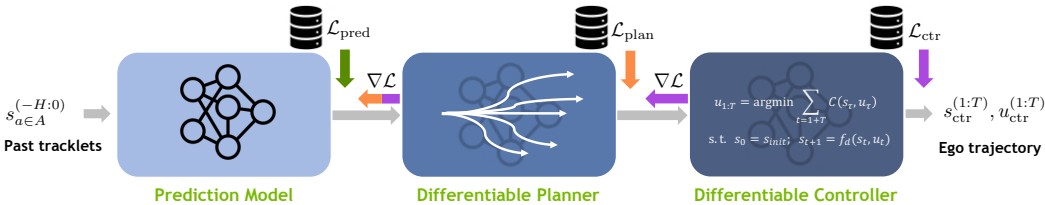

Figure 2: DiffStack is comprised of differentiable modules for prediction, planning, and control. Importantly, this means that gradients can propagate backwards all the way from the final planning objective, allowing upstream predictions to be optimized with respect to downstream decision making.

downside of this approach is that all components are neural networks, preventing the incorporation of principled planning and control algorithms, which are otherwise common in practice.

**Differentiable algorithms.** The idea of making existing model-based algorithms differentiable opens up new design paradigms that blend data-driven deep learning with classic model-based algorithms [15]. When both neural network models and fixed algorithmic computation can be expressed in the same "neural programming" framework, one can perform joint end-to-end learning even in structured modular architectures. Differentiable algorithms have been explored in various robotic domains including localization [33, 34], mapping [35, 36], navigation [37, 38, 15], and robot control [39, 16, 40, 41]. Our work is motivated by the initial successes of these approaches in rather simple robotic domains. To the best of our knowledge, this is the first work to apply differentiable algorithms in the context of autonomous driving stacks including prediction, planning, and control.

# 3 DiffStack: A Differentiable and Modular Autonomous Vehicle Stack

DiffStack is an autonomous driving stack with modules for prediction, planning, and control (Fig. 2). At a high level, DiffStack produces a trajectory for the AV to follow given a goal state, the tracked states of other agents, and a lane graph. To do so, it predicts where non-AV agents could move in the future; uses the predictions to select a safe, low-cost, and dynamically-feasible motion plan from a set of candidates; and finally optimizes the plan to yield a continuous and smooth ego-trajectory.

The key feature of DiffStack is that it is *differentiable*, i.e., we can compute $\partial \mathcal{L}_j / \partial \theta_i$, the gradient of the training objective $\mathcal{L}_j$ of downstream module $j$ with respect to the parameters $\theta_i$ of upstream module $i$. Differentiability enables training all parameters jointly for the overall objective of the stack with a gradient-based optimizer. We can also interpret DiffStack as a neural network with interpretable internal representations and fixed computational blocks for planning and control. Section 3.1 will describe how these components are made differentiable.

The main use case considered in this paper is to train the prediction module with respect to a final control objective. Different agents' predictions have different importance to the ego-vehicle (see examples in Appendix B). Standard metrics used to train prediction models are agnostic to the consequences of downstream planning, treating harmless and dangerous errors equally [26, 27]. By training directly for the end-objective of ego control, we expect DiffStack's prediction module to not only be accurate, but also synergize with planning and yield better overall ego controls.

We consider two open-loop training settings. In reinforcement learning (RL) experiments we optimize a hindsight cost that measures the quality of an ego trajectory in *hindsight*, i.e., after observing the future trajectory of other agents in the scene. The hindsight cost can be viewed as a negative reward for RL. In imitation learning (IL) experiments we compare the planned ego trajectory to the ego-vehicle's enacted motion in the data through a mean-squared-error (MSE) loss. We additionally explore using DiffStack to tune the hand-specified planning and control cost functions. In this section, we describe each module in DiffStack and their respective training setups, with more details in Appendix C.

**Nomenclature.** *Ego* refers to the AV and *agent* is a non-AV vehicle or pedestrian. States, $s$, consist of 2D position, heading, and longitudinal velocity. Control variables, $u$, are heading rate and longitudinal acceleration. A trajectory is a sequence of states, or states and controls, depending on the context. Distance between states is measured by the 2D Euclidean distance, whereas root-mean-squared state distance over time is used for trajectories. We denote trainable parameters of the prediction module by $\theta$, and trainable parameters of the planning and control cost by $w$. We perform two types of optimization: *training* (also called *learning*) optimizes $\theta$ and $w$ to minimize a loss $\mathcal{L}$ over some data; *planning* and *control* optimizes the ego trajectory to minimize a cost function with $\theta$ and $w$ fixed.

### 3.1 DiffStack modules

**Prediction.** We employ Trajectron++ [42], a state-of-the art CVAE that takes $H$ seconds of state history for all agents as input, and outputs multimodal trajectory predictions for one agent $a \in A$,

$$\hat{s}_a^{(1:T)}(\theta) = \{\hat{s}_{a,k}^{(1:T)}(\theta)\}_{k \in K} = \text{CVAE}\left(s_{a' \in A}^{(-H:0)}; \theta\right), \tag{1}$$

where $k \in K$ is the mode of the output distribution. We will use $\hat{s}_a = \hat{s}_a^{(1:T)}(\theta)$ for brevity. The encoder of the CVAE processes agent state histories with recurrent LSTM networks and models inter-agent interactions using graph-based attention. The decoder is a GRU that outputs a Gaussian Mixture Model (GMM) for each future timestep. The GMM modes correspond to the CVAE's $K = 25$ discrete latent states. To ensure predictions are dynamically-feasible, GMMs are defined over controls and then integrated through a known (differentiable) dynamics function to produce a trajectory. We use the default model configuration without map and ego conditioning. We augment the input states with an ego-indicator variable to allow for ego-agent relation reasoning. The raw prediction training objective is the InfoVAE loss, $\mathcal{L}_{\text{pred}} = \mathcal{L}_{\text{InfoVAE}}\left(\hat{s}_a, s_a^{\text{gt}}\right)$, the same as for the original Trajectron++.

**Planning.** The planner is a sampling-based algorithm that generates a set of $N$ dynamically-feasible ego trajectory candidates, $\mathcal{P} = \{s_n, u_n\}_{n \in N}$, and selects the candidate with the lowest cost. Namely,

$$s_{\text{plan}}, u_{\text{plan}} = \underset{s_n, u_n \in \mathcal{P}}{\arg\min} \, C(s_n, u_n; \hat{s}_{a \in A}, g, m; w), \tag{2}$$

where $C$ is the cost function, $\hat{s}_{a \in A}$ are multimodal predictions for all agents from the prediction module, $g$ is a given goal, and $m$ is a lane graph. To generate trajectory candidates we sample a set of lane-centric terminal states, fit a cubic spline from the current state to the terminal state, and reject dynamically-infeasible trajectories. The cost function is a weighted sum of handcrafted terms,

$$C(\cdot; w) = w_1 C_{\text{coll}}(s, \hat{s}_{a \in A}) + w_2 C_{\text{g}}(s, g) + w_3 C_{\ell \perp}(s, m) + w_4 C_{\ell \angle}(s, m) + w_5 C_u(u), \tag{3}$$

penalizing collisions, distance to the goal, lateral lane deviation, lane heading deviation, and control effort, respectively. Most notably, the collision term incorporates predictions into planning, $C_{\text{coll}}(s, \hat{s}_{a \in A}) = \sum_{a \in A} \sum_{t \in 1:T} \varphi\left(\sum_{k \in K} \pi_k ||s^{(t)} - \hat{s}_{a,k}^{(t)}||^2\right)$, where $\hat{s}_{a,k}$ is the $k$-th mode of the predicted trajectory distribution for agent $a$, $\pi_k$ is the probability of the $k$-th mode, $\varphi$ is a Gaussian radial basis function, and $||\cdot||$ is the Euclidean norm. Without loss of generality, in experiments we only do prediction for the vehicle closest to ego, and use GT futures for other agents. The remaining terms of (3) are defined in Appendix C.

We make the planner differentiable by relaxing the $\arg\min$ operator of (2) into sampling from a categorical distribution, and choosing a cross entropy loss for $\mathcal{L}_{\text{plan}}$, effectively treating the planner as a classifier during training. Specifically, we define the probability $p_n$ of choosing candidate $n$ as $p_n = \exp(-\beta C(s_n, u_n; \cdot; w))/Z$, where $\beta$ is a temperature parameter and $Z$ is the normalization constant. The planning loss is then $\mathcal{L}_{\text{plan}} = \mathcal{L}_{\text{CE}} = -\sum_{n \in N} \mathbb{1}(s_n = s^*) \log p_n$, where $s^*$ is a target trajectory chosen depending on the training setup (see Section 3.2). We can now compute $\frac{\delta \mathcal{L}_{\text{plan}}}{\delta w} = \frac{\delta \mathcal{L}_{\text{CE}}}{\delta p_n} \frac{\delta p_n}{\delta C} \frac{\delta C}{\delta w}$; and similarly, $\frac{\delta \mathcal{L}_{\text{plan}}}{\delta \theta} = \frac{\delta \mathcal{L}_{\text{CE}}}{\delta p_n} \frac{\delta p_n}{\delta C} \frac{\delta C}{\delta C_{\text{coll}}} \frac{\hat{s}_a}{\theta}$, where all terms exist.

**Control.** The control module performs MPC over a finite horizon using an iterative box-constrained linear quadratic regulator (LQR) algorithm [43]. Formally, we aim to solve

$$s_{\text{ctr}}, u_{\text{ctr}} = \underset{s,u}{\arg\min} \, C(s, u; \hat{s}_{a \in A}, g, m; w) \text{ s.t. } s^{(0)} = s^{\text{init}}, \, s^{(t+1)} = f_{\text{d}}(s^{(t)}, u^{(t)}), \, \underline{u} \leq u \leq \overline{u}, \tag{4}$$

where $C$ denotes the cost function, $f_{\text{d}}$ the dynamics, $s^{\text{init}}$ the current ego state, and $\underline{u}, \overline{u}$ the control limits. We use the cost defined in (3) for $C$ and the dynamically-extended unicycle [44] for $f_{\text{d}}$. We initialize the trajectory with $u_{\text{plan}}$ from the planner. The algorithm then iteratively forms and solves a quadratic LQR approximation of (4) around the current solution $s^{(i)}, u^{(i)}$ for iteration $i$, using first- and second-order Taylor approximations of $f_{\text{d}}$ and $C$, respectively. The trajectory is updated to be close to the LQR optimal control while also decreasing the original non-quadratic cost. We stop iterations upon convergence or a fixed limit.

To make the control algorithm differentiable we leverage Amos et al. [16]. The iLQR optimal trajectory, $s_{\text{ctr}}$, can be differentiated wrt. $C$ and $f_{\text{d}}$ by implicitly differentiating the underlying KKT conditions of the last LQR approximation. The gradients can be analytically computed by one additional backward pass of a modified iterative LQR solver. If iLQR fails to converge, we do not backpropagate gradients. In our setting $f_{\text{d}}$ is fixed. We compute gradients wrt. cost parameters, $\frac{\delta \mathcal{L}_{\text{ctr}}}{\delta w} = \frac{\delta \mathcal{L}_{\text{ctr}}}{\delta s_{\text{ctr}}} \frac{\delta s_{\text{ctr}}}{\delta C} \frac{\delta C}{\delta w}$, and further wrt. prediction model parameters, $\frac{\delta \mathcal{L}_{\text{ctr}}}{\delta \theta} = \frac{\delta \mathcal{L}_{\text{ctr}}}{\delta s_{\text{ctr}}} \frac{\delta s_{\text{ctr}}}{\delta C} \frac{\delta C}{\delta C_{\text{coll}}} \frac{\hat{s}_a}{\theta}$, where $\mathcal{L}_{\text{ctr}}$ is the control loss, which we will discuss next.

Table 1: Open-loop evaluation results. DiffStack outperforms all baselines in planning and control metrics because it trains its prediction model for the downstream task.

| Method | Pred. Obj. | End Obj. | ADE (m) ↓ ± Std. Err. | NLL (nats) ↓ ± Std. Err. | Planning Loss ↓ $\times 10^{-2}$, ± Std. Err. | Hindsight Cost ↓ $\times 10^{-2}$, ± Std. Err. |
|---|---|---|---|---|---|---|
| No prediction | | | – | – | $0.00 \pm 0.00$ | $0.00 \pm 0.00$ |
| Standard | ✓ | | $1.32 \pm 0.06$ | $0.42 \pm 0.04$ | $-3.76 \pm 0.23$ | $-1.61 \pm 0.05$ |
| Distance weighted | ✓ | | $1.34 \pm 0.06$ | $1.19 \pm 0.10$ | $-4.04 \pm 0.22$ | $-1.68 \pm 0.04$ |
| ∇Cost weighted [27] | ✓ | | $1.43 \pm 0.10$ | $-\mathbf{1.06 \pm 1.52}$ | $-4.29 \pm 0.20$ | $-1.77 \pm 0.07$ |
| DiffStack (no $\mathcal{L}_{\text{pred}}$) | | ✓ | $1.37 \pm 0.06$ | $8.01 \pm 0.75$ | $\mathbf{-5.13 \pm 0.22}$ | $\mathbf{-1.87 \pm 0.04}$ |
| DiffStack | ✓ | ✓ | $\mathbf{1.27 \pm 0.07}$ | $0.38 \pm 0.05$ | $\mathbf{-5.13 \pm 0.18}$ | $-1.86 \pm 0.04$ |
| GT prediction | | | – | – | $-7.56 \pm 0.00$ | $-2.25 \pm 0.00$ |

## 3.2 End-to-end training

An important question for data-driven AV stacks is the training objective and data. Learning in the real world is prohibitively expensive, and building a simulator with realistic traffic agent behavior is an open challenge. Accordingly, standard practice is to perform open-loop training with human driving data [45, 13, 31, 23, 24, 46, 47]. We consider two common types of open-loop training settings: reinforcement learning (RL) and imitation learning (IL).

In the RL setting, we aim to minimize the (hindsight) cost of output ego trajectories, $s_{\text{ctr}}, u_{\text{ctr}}$, over a dataset, $\mathcal{L}_{\text{ctr}} = \mathcal{L}_{\text{HC}} = C_H \left( s_{\text{ctr}}, u_{\text{ctr}}; s_{a \in A}^{\text{gt}}, g, m; w \right)$. The hindsight cost $C_H$ captures the quality of a trajectory in *hindsight*, i.e., after knowing the future trajectory of non-ego agents $s_{a \in A}^{\text{gt}}$, similar to the concept of rewards in RL. We choose $C_H$ identical to the control cost $C$ defined in (3), but with GT future trajectory inputs instead of predictions, and fixed $w$. Note that our open-loop training setup does not account for the effect of ego actions on other agents; nevertheless, we treat recorded trajectories as GT futures, as in the prediction literature [48]. In this setting we only train the prediction model. Given GT futures $\frac{\delta \mathcal{L}_{\text{HC}}}{\delta \theta} = \frac{\delta \mathcal{L}_{\text{HC}}}{\delta \{s_{\text{ctr}}, u_{\text{ctr}}\}} \frac{\delta \{s_{\text{ctr}}, u_{\text{ctr}}\}}{\delta \theta}$ where both terms exists. For the planner's target we choose the trajectory candidate with the lowest hindsight cost, $s^* = \arg \min_{s_n, u_n \in \mathcal{P}} C_H(s_n, u_n; \cdot)$.

In the IL setting we aim to minimize the MSE between the output ego trajectory and the GT in the dataset, $\mathcal{L}_{\text{ctr}} = \mathcal{L}_{\text{IL}} = \sum_{t=1:T} ||s_{\text{ctr}}^{(t)} - s^{\text{gt},(t)}||^2$. For the planner's target we choose $s^* = s^{\text{gt}}$.

The total loss for training DiffStack is $\mathcal{L} = \alpha_1 \mathcal{L}_{\text{pred}} + \alpha_2 \mathcal{L}_{\text{plan}} + \alpha_3 \mathcal{L}_{\text{ctr}}$. We experiment with different $\alpha_i$ values, including setting each $\alpha_i$ to zero. In the following we omit $\alpha_i$ for brevity.

## 4 Experiments

We ask the following questions: 1) can DiffStack learn predictions that lead to better plans? 2) how does DiffStack compare to alternative planning-aware training techniques? 3) can DiffStack correct integration errors? 4) can DiffStack learn a cost from imitation? 5) do our results translate to closed-loop simulation? We begin with open-loop experiments (1–4), and then present closed-loop results (5) in Section 4.3.

### 4.1 Experimental setup

**Dataset.** We use the nuScenes dataset [17], comprised of state annotations for vehicles and pedestrians in 1000 scenes across Boston and Singapore. For training and open-loop evaluation we sample suitable planning scenarios from the dataset with $H = 4$s history and $T = 3$s future data. For each scenario, we choose one vehicle to act as the ego. Details including dataset splits are in Appendix D.

**Metrics.** We evaluate DiffStack's prediction module via standard prediction metrics: average displacement error (ADE) for the most-likely prediction and negative log-likelihood (NLL) for its full distributional output. To evaluate planning and control, we use the cross-entropy planning loss $\mathcal{L}_{\text{plan}} = \mathcal{L}_{\text{CE}}$, and control loss $\mathcal{L}_{\text{ctr}}$. We report the hindsight cost for reinforcement learning ($\mathcal{L}_{\text{ctr}} = \mathcal{L}_{\text{HC}}$) and MSE for imitation learning experiments ($\mathcal{L}_{\text{ctr}} = \mathcal{L}_{\text{MSE}}$). Since control is an AV stack's end-goal, $\mathcal{L}_{\text{ctr}}$ reflects the overall performance of the stack. To make these metric values more interpretable, we report them relative to a **No prediction** baseline and a **GT prediction**-based oracle. The baseline ego plans without predictions (ignoring the predicted agent), providing a performance lower bound. The GT oracle plans with GT futures in place of predictions, providing a notion of a

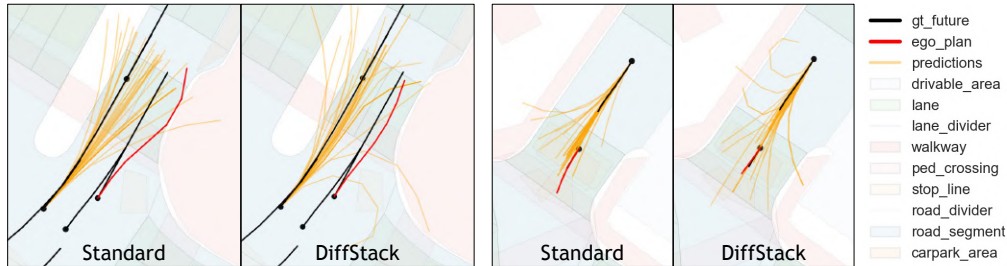

Figure 3: **Left**: The standard stack predicts an unlikely lane intrusion with high confidence, causing the planner to swerve. In contrast, DiffStack's prediction assigns most of the probability mass to reasonable trajectories that consider the presence of the ego vehicle, and the resulting motion plan stays well within lane boundaries. **Right**: The standard stack incorrectly predicts that a vehicle would collide into the ego-vehicle from behind with high probability, causing the ego-vehicle to move into the intersection. DiffStack assigns more probability to the agent slowing down, as a result the ego-vehicle slows down before the intersection.

performance upper bound. All results are averaged over our validation set, and we report standard errors of the mean over 5 training seeds.

**Baselines.** We compare DiffStack with standard AV stacks. For a fair comparison, we use the same set of modules as in DiffStack, but without making the planner and controller differentiable. We only train the prediction model, using increasingly planning-aware training objectives. **Standard** trains the prediction model with only the prediction loss $\mathcal{L}_{\mathrm{pred}}$, unaware of downstream planning. Next, we re-weight prediction losses (in each batch) based on a handcrafted measure of relevance for planning. **Distance weighted** weights losses with the inverse distance between the ego and agent GT futures. $\nabla$**Cost weighted** uses the magnitude of the control cost gradient with respect to the GT future trajectory of the agent, proposed as a planning-aware prediction metric in [27]. Finally, DiffStack backpropagates gradients of the final loss to the prediction module. **DiffStack (no $\mathcal{L}_{\mathrm{pred}}$)** uses $\mathcal{L} = \mathcal{L}_{\mathrm{plan}} + \mathcal{L}_{\mathrm{ctr}}$; **DiffStack** uses $\mathcal{L} = \mathcal{L}_{\mathrm{pred}} + \mathcal{L}_{\mathrm{plan}} + \mathcal{L}_{\mathrm{ctr}}$.

**Implementation details.** We implement DiffStack in PyTorch [49] and build on the open-source code of Trajectron++ [42] and Differentiable MPC [16]. We use $H = 4$s, $T = 3$s, and $\Delta t = 0.5$s. We train all models for 20 epochs using 4 NVIDIA Tesla V100 GPUs, taking 10–20 hours. Additional details are in Appendix C. The code is available at `https://sites.google.com/view/diffstack`.

### 4.2 Open-loop results

**End-to-end training is useful.** Our main results are summarized in Table 1. The key observation is that DiffStack improves the effectiveness of predictions on ego planning by up to 15.5% compared to standard training ($\mathcal{L}_{\mathrm{ctr}} = -1.86$ vs. $-1.61$); and reduces the gap to the cost attainable with a GT-informed oracle by 39.1% ($-1.86 + 2.25$ vs. $-1.68 + 2.25$). DiffStack also improves planning without significantly impacting raw prediction accuracy (ADE=1.27 vs. 1.32, within standard error). Alternative methods that re-weight the prediction loss in a planning-aware manner (rows 2 and 3) also help, but less than DiffStack. Surprisingly, we can even recover comparable prediction performance in terms of ADE when training solely for planning and control objectives (row 4). The poor distribution fit (high NLL) is due to our control cost being agnostic to the variance of the predicted GMMs.

**DiffStack makes fewer prediction errors that affect planning.** Qualitatively analysing predictions and plans shows that the main source of improvement from end-to-end training is the reduction of spurious and/or unrealistic predictions that lead to plans with large (unnecessary) deviations from the lane center or the goal position. Fig. 3 shows two particular examples. In both cases, DiffStack's predictions are more realistic and accurate, yielding much more reasonable downstream plans.

**DiffStack can compensate for system integration errors.** Integrating independently developed modules into an AV stack can be challenging due to, e.g., misaligned module interfaces. Table 2 shows results for an experiment that explores this issue. We introduce an artificial interface mismatch between prediction and planning by adding a fixed 1m offset to all GT prediction targets. As a result, the standard model yields very poor plans (note the positive value in row 1, indicating worse performance than baseline planning). However, DiffStack can compensate for interface mismatch and substantially improves the overall plans, suggesting that it can learn to correct erroneous predictions that affect planning. More broadly, these results demonstrate the potential for differentiable AV stacks

Table 2: System integration results. DiffStack compensates for errors in module integration. Even if GT prediction targets are erroneously offset by 1m, training with downstream losses corrects the prediction model's bias and improves overall system performance.

| Method | Pred. Obj. | End Obj. | ADE (m) ↓ ± Std. Err. | NLL (nats) ↓ ± Std. Err. | Planning Loss ↓ ×10⁻², ± Std. Err. | Hindsight Cost ↓ ×10⁻², ± Std. Err. |
|---|---|---|---|---|---|---|
| No prediction | | | – | – | $0.00 \pm 0.00$ | $0.00 \pm 0.00$ |
| Standard-biased | ✓-biased | | $3.64 \pm 0.34$ | $8.43 \pm 0.72$ | $16.22 \pm 2.73$ | $3.13 \pm 0.40$ |
| DiffStack-biased | ✓-biased | ✓ | $\mathbf{2.01 \pm 0.08}$ | $\mathbf{4.20 \pm 0.09}$ | $\mathbf{-3.58 \pm 0.96}$ | $\mathbf{-1.61 \pm 0.12}$ |
| GT prediction | | | – | – | $-7.56 \pm 0.00$ | $-2.25 \pm 0.00$ |

Table 3: Imitation learning results. DiffStack's plans better match the GT ego trajectories from data.

| Method | Pred. Obj. | End Obj. | ADE (m) ↓ ± Std. Err. | NLL (nats) ↓ ± Std. Err. | Planning Loss ↓ ×10⁻³, ± Std. Err. | MSE (m²) ↓ ×10⁻², ± Std. Err. |
|---|---|---|---|---|---|---|
| No prediction | | | – | – | $0.00 \pm 0.00$ | $0.00 \pm 0.00$ |
| Standard | ✓ | | $\mathbf{1.25 \pm 0.02}$ | $0.53 \pm 0.23$ | $-0.50 \pm 0.05$ | $0.60 \pm 0.15$ |
| DiffStack | ✓ | ✓ | $1.43 \pm 0.14$ | $\mathbf{0.30 \pm 0.23}$ | $\mathbf{-0.80 \pm 0.04}$ | $\mathbf{-1.80 \pm 0.16}$ |
| GT prediction | | | – | – | $-1.10 \pm 0.00$ | $-2.80 \pm 0.00$ |

to reduce various development/engineering costs associated with developing AVs, e.g., by replacing tedious manual parameter tuning with end-to-end data-driven optimization. We further explore this topic in the cost tuning experiments below.

**DiffStack can imitate humans better.** Imitation learning results are summarized in Table 3. The overall performance of the stack is now measured by $\mathcal{L}_{\mathrm{ctr}} = \mathcal{L}_{\mathrm{MSE}}$, the MSE between planned and human expert trajectories relative to the MSE for the baseline planner (without predictions). We observe a similar trend as before; however, the results also reveal some limitations. The standard stack performs worse in terms of MSE than the baseline stack which neglects predictions entirely (note the positive sign for relative $\mathcal{L}_{\mathrm{MSE}}$ in row 1). While DiffStack improves MSE significantly, the absolute difference is small. We hypothesize this is caused by our cost function not being rich enough to fully capture human driving behavior; and because agent-agent interactions that affect planning are rare in nominal driving data (comprised mainly of lane- and speed-keeping). These limitations could be addressed by a richer cost function and alternative goal definitions, which we leave to future work.

**DiffStack can learn a better control cost.** While our main use case in this paper is to learn planning-aware prediction, DiffStack opens up various other opportunities for data-driven optimization of various components of the AV stack. For example, an important challenge in practice is to design a cost function

Table 4: DiffStack learns a better control cost.

| Method | Planning Loss ↓ | MSE (m²) ↓ |
|---|---|---|
| Hand-tuned | $2.67 \pm 0.00$ | $0.38 \pm 0.00$ |
| Learned | $\mathbf{2.41 \pm 0.01}$ | $\mathbf{0.32 \pm 0.00}$ |

for planning and control. In this experiment, we explore DiffStack's potential to learn interpretable control costs that optimize the quality of output plans from data. Table 4 reports results in the imitation learning setting, where we first train a prediction module (as before), then we train for an additional 20 epochs allowing DiffStack to update the weights $w_i$ of the control cost (3) by backpropagating gradients from the final control loss $\mathcal{L}_{\mathrm{ctr}}$. Compared to the default hand-tuned weights (row 1), DiffStack significantly decreases plan MSEs (row 2). The resulting learned weights $w_i$ are lower for the control effort term, and higher for the goal and lane keeping terms (see Appendix A). To ensure safety, we fix the weights for collision avoidance. We leave comparison with alternative techniques for learning control costs to future work. We expect DiffStack to be more effective compared to, e.g., Bayesian optimization, where the number of learned parameters is large.

**Ablations.** Results of an ablation study can be found in Appendix A. In short, improvements from DiffStack are consistent with our observations when training for $\mathcal{L} = \mathcal{L}_{\mathrm{pred}} + \mathcal{L}_{\mathrm{plan}}$; $\mathcal{L} = \mathcal{L}_{\mathrm{pred}} + \mathcal{L}_{\mathrm{ctr}}$; when changing the relative scale of loss components; with higher time resolution $\Delta t = 0.1$ for planning and control; and when only using one of the planning or control modules.

### 4.3 Closed-loop evaluation

**Simulation.** We perform closed-loop simulation using a simple log-replay setup, where ego states are unrolled based on the planned control outputs and known dynamics, while non-ego agents follow their fixed trajectories recorded in the dataset. The evaluation scenarios are $T_{\mathrm{sim}} = 10s$ long. The goal

Table 5: Closed-loop results. DiffStack outperforms the standard stack similarly to open-loop results.

| Method | Trajectory Cost $\times 10^{-2} \downarrow$ | Open-loop Cost $\times 10^{-2} \downarrow$ | Collision Cost $\times 10^{-2} \downarrow$ | Lane Keep Cost $\times 10^{-2} \downarrow$ | Control Effort $\times 10^{-2} \downarrow$ | Deviation (m) $\times 10^{-2} \downarrow$ |
|---|---|---|---|---|---|---|
| No prediction | 0.00 | 0.00 | 0.00 | 0.00 | 0.00 | 0.00 |
| Standard | 0.87 | 0.33 | $-\mathbf{8.21}$ | 8.91 | 0.18 | 1.05 |
| DiffStack | $-\mathbf{4.37}$ | $-\mathbf{2.40}$ | $-8.20$ | $\mathbf{4.84}$ | $-\mathbf{0.99}$ | $\mathbf{0.98}$ |
| GT prediction | $-6.30$ | $-2.58$ | $-8.54$ | 3.34 | $-1.08$ | 0.70 |

and lane inputs for planning are updated in each simulation step based on the logged ego trajectory. We compute the following metrics. **Trajectory Cost** captures closed-loop performance: it evaluates the cost function $C$ on unrolled simulation trajectories, $1/T_{\text{sim}} \sum_{t=1:T_{\text{sim}}} C(s_{\text{sim}}^{(t)}, u_{\text{sim}}^{(t)}; s_{a \in A}^{\text{gt},t}, m^{(t)})$, where $s_{\text{sim}}$ and $u_{\text{sim}}$ are the unrolled ego state and control. We exclude the goal cost term because the goal is updated in each simulation step. **Open-loop Cost** is the average of hindsight costs calculated open-loop at each simulation step, $\mathcal{L}_{\text{HC}} = C(s_{\text{ctr}}, u_{\text{ctr}}; s_{a \in A}^{\text{gt}}, g, m)$. **Collision Cost** is the collision term in the trajectory cost, $w_1 C_{\text{coll}}(s_{\text{sim}}^{(t)}, s_{a \in A}^{\text{gt},(t)})$. **Lane Cost** is the sum of lane keeping terms, $w_3 C_{\ell \perp} + w_4 C_{\ell \angle}$. **Control Effort** is the control effort term, $w_5 C_u$. **Deviation** is the average distance between unrolled and logged ego trajectories from the data, $||s_{\text{sim}}^{(t)} - s^{\text{gt},(t)}||$.

**Results.** Simulation results are in Table 5. Most importantly, we observe similar relative improvement for DiffStack in terms of closed loop trajectory cost as in terms of open-loop cost. DiffStack performs substantially better that the *no prediction* baseline, but somewhat surprisingly the standard stack performs worse, both in terms of closed-loop trajectory cost and open-loop cost. Analysing the cost components sheds light on possible reasons. As expected, incorporating predictions into planning results in lower collision costs, but higher lane keeping costs. The contribution of these two terms to the average cost depends on the data distribution, e.g., the frequency of close interactions where predictions are useful. As we saw earlier in Fig. 3, poor predictions in the standard stack frequently lead to unnecessary lane deviations in the planner. This is reflected in the substantially higher lane cost and control effort cost for the standard stack compared to DiffStack in Table 5.

## 5 Limitations & Conclusions

**Limitations.** One limitation of our work is the open-loop training and log-replay based simulation setup. In lieu of a real-world AV or a simulator with strong behavioral realism, this is standard practice; however, recent efforts on accurate behavior simulation could be leveraged in the future [50, 51, 52, 53, 54]. Our implementation of DiffStack also has limitations. First, it is not differentiable wrt. *all* possible parameters, e.g., no gradients flow from the control loss to the planner's trajectory candidate generator. Future work may develop more sophisticated differentiable algorithms and explore ideas for gradient approximation for non-differentiable components [55]. Second, individual modules could be improved, e.g., by adding a more sophisticated prediction model, improving candidate sampling in the planner, and adding trust-region constraints to the controller. Finally, even though hand-engineered components, modularity, and intermediate training objectives in differentiable stacks remedy challenges of learning AV policies end-to-end, other challenges naturally remain. For instance, designing an overall performance metric, or reducing the scarcity of and cost to obtain (interesting) driving data remain open problems, each of which are impactful areas of future work.

**Conclusions.** In this paper we take a step towards fully-differentiable and modular AV stacks by introducing key differentiable components for prediction, planning, and control. Our experimental results show the potential benefits of jointly training modules of AV stacks for downstream performance. For motion prediction in particular, our results indicate that there is value in moving from purely prediction-oriented evaluation metrics towards downstream task-oriented metrics, in line with arguments in recent work [26, 27, 28]. While our experiments focused on learning planning-aware predictions, DiffStack opens up various exciting opportunities for task-oriented learning in modular stacks. For example, we may learn a rich neural network as part of the control cost to capture hard-to-engineer concepts, e.g., reasoning with occlusions or accounting for uncertainties. Eventually, we envision having differentiable modules for the entire AV stack, allowing any subset of modules to be learned and optimized for a downstream task. Overall, this would relax information bottlenecks and enable uncertainty to more easily propagate through the stack without needing to forgo interpretability, modularity, and verifiabilty of the various components.

## Acknowledgments

We thank Yuxiao Chen and Edward Schmerling for generous help with our planner implementation and closed-loop evaluation experiment; Christopher Maes for fruitful discussions on differentiable MPC and cost learning experiments; and Nikolai Smolyanskiy for valuable high-level feedback.

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
