# OpenReview forum: "DiffStack: A Differentiable and Modular Control Stack for Autonomous Vehicles"
_robot-learning.org/CoRL/2022/Conference — CoRL 2022 Poster_

### Official Review · Reviewer_a5FX · 2022-07-28

**Originality:** Very Good
**Technical Quality:** Excellent
**Clarity Of Presentation:** Excellent
**Impact:** 4

**Recommendation:**

Strong Accept: I recommend accepting the paper and will argue for my recommendation even if other reviewers hold a different opinion.

**Summary:**

The authors tackle the task of offering a full autonomous driving software stack. In contrast to classical approaches that focus on modular software containing of perception, planning and control, the authors present here software components that are full differentiable. The authors focus on prediction, planning and control modules that are made differentiable, train it in a simulation environment (nuScenes) and then show the results in comparison to baselines. The authors display that this approach - in contrast to prior End-to-End methods - gives more modularity and interpretability.

**Issues:**

The biggest issue i have with the paper are the results and explanations of the outcome and why diffstack is better. Since the baseline is also diffstack but trained with different objectives (correct?) I do not see the justification for the discussion. The discussion section is great and shows why Diffstack is good, but I miss the comparison to a classical stack or a real end-to-end pipeline

**Quality Of The Limitations Section:**

Additional details required

**Reviewer Expertise:**

5: The reviewer is absolutely certain that the evaluation is correct and very familiar with the relevant literature

**Robotics Focus:**

Highly relevant to robotics but no hardware experiments

**Strengths And Weaknesses:**

Strengths:

-The paper is well-written and good to read, the technical explanations are exceptional
- The paper tackles a very hot topic in the field, especially for AVs were we are currently seeing bottlenecks in hand-engineered algorithms
- Honest opinion about the limitations of this approach, it is not a full full differentiable stack
- The results indicate that "there is value in moving from purely prediction-oriented evaluation metrics towards downstream task-oriented metrics" -> This was new to me and is a significant result, especially with all the work in prediction-algorithms right now

Weaknesses:
- No hardware experiments were provided
- No Training results were provided. Although it is stated that the nuScens datasets is used there is nothing explained about training quality and training time
- No comparison to a modular software stack (classical pipeline) or a complete end-to-end pipeline. I do not understand the comparison with the "baseline" (We compare DiffStack to the same prediction-planning-control stack, but with modules
219 trained for different objectives that capture a varying degree of planning-awareness) ??
- The authors claim with the title its a full-stack but in the limitations, they show that this approach is only performed in open-loop training and evaluation

**Summary Of Recommendation:**

For me this paper is a strong accept since the paper contains many components and ideas that will potentially have a major impact in the field of autonomous vehicles. The paper is both well-written and technically deep, the authors explain whats going on in their different Software Stack Modules (Predction, Planning, Control) and how they are training in and end-to-end fashion. Generally that paper contains all important components and is of high interest for the community

---

> ### Author Response · Authors · 2022-08-24
> **Response to review**
>
> Thank you for the careful evaluation and encouraging feedback! We respond to your questions/comments below.
>
> - **Comparison to a modular software stack / end-to-end pipeline.**
> Please refer to our joint response above and the new results in the attached pdf.
> To specifically respond to your concern, the baseline “Standard” does represent a standard modular stack. To make the comparison fair, both the standard stack and DiffStack use the standard/differentiable equivalent of the same prediction, planning, and control modules, but only DiffStack can backpropagate gradients through modules. The two other baselines (“Distance weighted” and “GradCost weighted'') are also standard modular stacks, but here we train their prediction model with an alternative, weighted prediction loss that captures a notion of importance for planning. We will make this more clear in the final text in the revised version of the paper.
>
> - **Only open-loop results.**
> Please refer to our joint response above and the new results in the attached pdf.
> In short, we added results for a simple closed-loop simulation experiment, and found that the observed benefits of DiffStack over the standard stack translate to our closed-loop evaluation setting.
>
> - **Training details.**
> Due to space constraints we have not included training details in the main text, but some details on the training process are provided in Appendix D.

---

> > ### Comment · Reviewer_a5FX · 2022-08-26
> > **Thank you**
> >
> > Thank you very much for your answers.

---

### Official Review · Reviewer_oNtX · 2022-07-29

**Originality:** Good
**Technical Quality:** Very Good
**Clarity Of Presentation:** Very Good
**Impact:** 3

**Recommendation:**

Weak Accept: I recommend accepting the paper, but will not argue for my recommendation if the majority of other reviewers have a different opinion.

**Summary:**

This paper presents an end-to-end differentiable stack for autonomous driving. The prediction module is a neural network, and the planning and control modules are hand-designed algorithms. Notably, the hand-designed algorithms are differentiable, which allows training of the upstream prediction module for a downstream control objective by backpropagating gradients through these hand-designed algorithms. The authors make planning differentiable by replacing the argmin operation with sampling from a categorical distribution, and the authors make use of an off-the-shelf differentiable MPC algorithm to make control differentiable. The authors show open-loop, offline results which indicate that this kind of planning-aware training of the prediction module improves performance.


**Issues:**

N/A

**Quality Of The Limitations Section:**

Limitations are addressed clearly

**Reviewer Expertise:**

2: The reviewer is willing to defend the evaluation, but it is quite likely that the reviewer did not understand central parts of the paper

**Robotics Focus:**

Highly relevant to robotics but no hardware experiments

**Strengths And Weaknesses:**

Strengths
- A step towards end-to-end training of prediction modules, while retaining the interpretability / stability of hand-designed control and planning algorithms.
- The paper is well-written and easy to parse.

Weaknesses
- Only offline / open-loop evaluation, so it is difficult to predict what real world performance will be like.
- I found the metrics to be somewhat hard to interpret, but this could be because I’m not a researcher in the AV space.


**Summary Of Recommendation:**

I do not conduct research in autonomous driving, so take my recommendation with a grain of salt (this is the main reason I selected weak accept instead of strong accept). Overall, I felt the paper was well-motivated and executed, and tackles what I believe is an important problem. I recommend acceptance.

---

> ### Author Response · Authors · 2022-08-24
> **Response to review**
>
> Thank you for the careful evaluation and positive comments! We respond to your questions/comments below.
>
> - **Only open-loop evaluation.**
> Please refer to our joint response above and the new results in the attached pdf.
> In short, we added results for a simple closed-loop simulation experiment, and found that the observed benefits of DiffStack over the standard stack translate to our closed-loop evaluation setting.
>
> - **Metrics.**
> Thank you for raising this point, we will extend our explanation in the revised version of the paper. In short, ADE and NLL are standard metrics for the prediction module; Planning loss measures the quality of the planner’s output; and Hindsight cost measures the quality of the controller’s output (which is the overall system output). More specifically, ADE (average displacement error) measures the distance between the mean prediction and the GT future of a predicted agent. NLL is the negative log-likelihood of the GT agent future under the distribution output by the prediction model. Planning loss is the cross-entropy loss we used to train the planner (line 160), it is computed by treating the planner as a classifier over a set of trajectory candidates. Hindsight cost measures the quality of an output trajectory through a hand-crafted cost function (Eq. 3) given the GT future trajectories of non-ego agents.

---

> > ### Comment · Reviewer_oNtX · 2022-08-26
> > **Thanks**
> >
> > Thank you for your response.

---

### Official Review · Reviewer_jUVa · 2022-08-01

**Originality:** Good
**Technical Quality:** Fair
**Clarity Of Presentation:** Very Good
**Impact:** 3

**Recommendation:**

Weak Reject: I recommend rejecting the paper, but will not argue for my recommendation if the majority of other reviewers have a different opinion.

**Summary:**

Rather than building an AV stack in a modular fashion, this work introduces a differentiable and modular stack for prediction, planning and control. This enables optimization of upstream components such as prediction via backprop through planning and control.

**Issues:**

- The argument for a fully data-driven yet modular and interpretable AV architecture is compelling in general, but could be better articulated in this work. Particularly there is no comparison or demonstration of their approach providing a benefit  over a non-interpretable end-to-end system.  It would be great if the authors could find some way to quantify this, or at least qualitatively demonstrate the difference.



**Quality Of The Limitations Section:**

Limitations are addressed clearly

**Reviewer Expertise:**

3: The reviewer is fairly confident that the evaluation is correct

**Robotics Focus:**

Relevant but unlikely to deploy to hardware in near future

**Strengths And Weaknesses:**

Strengths:
- The method is still interpretable in the sense that it is composed of modules with specific purposes.
- The method is able to be trained end-to-end due to being fully differentiable.
- Comparison to a non-differentiable but modular method is made clearly.

Weakness:
- no real world or even simulator results.
- not differentiable to all possible parameters
- Comparison to an end-to-end neural network is not made. The authors note that not all parameters are differentiable. How does this system compare to an end-to-end neural network that doesn't attempt to provide an interpretability? How much performance is being sacrificed for interpretability? Can the authors provide quantitative or qualitative examples of how this interpretability is useful?





**Summary Of Recommendation:**

Paper is well written and well motivated.
The experimental section could be made much strong through additional baselines (particulary a non-modularized end-to-end system), and real world or even simulated experiments.

---

> ### Author Response · Authors · 2022-08-24
> **Response to review**
>
> Thank you for the careful evaluation and we greatly appreciate your constructive feedback! We respond to your questions/comments below.
>
> - **Real world / simulator results.**
> Please refer to our joint response above and the new results in the attached pdf.
> In short, we added results for a simple closed-loop simulation experiment, and found that the observed benefits of DiffStack over the standard stack translate to our closed-loop evaluation setting.
>
> - **Comparison to an end-to-end neural network.**
> Please refer to our joint response above. In short, we argue that it is generally difficult to compare modular and end-to-end learned approaches meaningfully as the results would be highly dependent on the exact architecture, training data, and evaluation scenarios. We did perform experiments with naive end-to-end policy networks, but did not obtain good performance.
>
> - **How much performance is being sacrificed for interpretability?**
> We hope that our response on comparison with an end-to-end neural network has partially addressed this question. In addition, we would like to point out that interpretability does not necessarily mean a sacrifice in performance. Beyond being interpretable, modular architectures (including DiffStack) encode numerous inductive biases which can improve generalization performance. Inductive biases can take the form of handcrafted representations (e.g. state), known transition dynamics, handcrafted planning cost, an explicit trajectory optimization algorithm, etc.
>
> - **Can the authors provide quantitative or qualitative examples of how this interpretability is useful?**
> We leverage the interpretable prediction outputs of DiffStack, among others, to perform the qualitative analysis in Figure 3, and compute prediction oriented quantitative metrics in Tables 1, 2, and 3.
> More generally the benefits of interpretable driving systems has been argued in various prior works, for example [13]. As discussed above, one benefit of interpretable modular architectures is their ability to encode inductive biases. Another benefit is reconfigurability: to deploy the stack on a new vehicle with different dynamics we only need to change the dynamics function; if the passenger of an AV has a stronger preference for comfort we can increase the corresponding weight in the planning cost function.
>
> - **Differentiability to all possible parameters.**
> Indeed our architecture is not differentiable w.r.t. all conceivable parameters. We discuss this issue in lines 292-295. However, our results show that our architecture still enables substantial gains over its non-differentiable counterpart. Future work may introduce new differentiable approximations or design choices to make other parameters trainable, for example, additional parameters of an interpretable planning cost. We note that standard neural networks are also not differentiable wrt. all (hyper-)parameters, e.g., the size parameter of a convolutional kernel; and gradients do not flow through all operations, e.g., the max-pool layer.

---

### Official Review · Reviewer_67Mn · 2022-08-04

**Originality:** Good
**Technical Quality:** Good
**Clarity Of Presentation:** Very Good
**Impact:** 3

**Recommendation:**

Weak Accept: I recommend accepting the paper, but will not argue for my recommendation if the majority of other reviewers have a different opinion.

**Summary:**

The authors present a fully differentiable yet modular pipeline for autonomous vehicle perception and control. It preserves the traditional abstractions such as perception, prediction, planning, and control but implements differentiable versions of these components so that gradients can pass all the way from task performance back to perception. They demonstrate the value of this on a trajectory prediction task.

**Issues:**

See above.

**Quality Of The Limitations Section:**

Limitations are addressed clearly

**Reviewer Expertise:**

3: The reviewer is fairly confident that the evaluation is correct

**Robotics Focus:**

Highly relevant to robotics but no hardware experiments

**Strengths And Weaknesses:**

Strengths:

 - The idea is well-motivated and the approach seems well executed. The approach preserves some of the important interpretability charateristics of traditional methods while still enabling end-to-end learning.
 - Using task performance to learn perceptual models is a promising area of research, and this paper showcases a possible method for doing so in the case of autonomous vehicles.


Weaknesses / Questions:

 - The authors should highlight to what degree the differentiable components are novel. To my understanding they have mostly taken existing components and assemble them together so the contribution should be mostly considered at the system level. If this is not the case the authors should clarify.

 - The cost function in (3) contains a lot of weights. How sensitive is the algorithm to the selection of these weights? Similar question for $\alpha_{1:3}$ in the combined loss function.

 - How is the discretization of $s$ chosen for the planner (i.e., what is the dimension of the Categorical distribution and why)?


**Summary Of Recommendation:**

Overall, the work presents a novel autonomous vehicle pipeline that is modular yet differentiable. Although none of the individual components seems to be particularly novel, the system that is constructed is and the advantages that it affords seem to be convincing.

---

> ### Author Response · Authors · 2022-08-24
> **Response to review**
>
> Thank you for the careful evaluation and positive feedback! We respond to your questions/comments below.
>
> - **Contribution.**
> Indeed our architecture utilizes existing components, but to the best of our knowledge we are the first to demonstrate that, through careful design choices and integration, these modules can be composed into an end-to-end trainable AV stack.
>
> - **Sensitivity to cost function parameters and the $\alpha_{1:3}$ loss weight parameters.**
> The cost function (and its parameters) naturally have an effect on the resulting plans, both for Standard stack and DiffStack. We expect DiffStack to improve over a standard stack with different cost function parameters as well. In the case of the $\alpha_{1:3}$ loss weights, we have investigated sensitivity to these parameters in the ablation study (Table 5 in the appendix, rows 6-7). The relative weight of the prediction loss vs. the downstream decision making losses behave as expected: the higher/lower the weight for the downstream planning and control losses the more/less planning and control metrics improve at the expense of slightly worse/better raw prediction metrics.
>
> - **Discretization of $s$ chosen for the planner.**
> We would like to clarify that the state $s$ is not discretized in the planner. The planner generates a set of $N$ trajectory candidates in continuous space. We form a categorical distribution over these continuous-space candidates for the purpose of training. The number of trajectory candidates is determined by the handcrafted generation process, which we detail in Appendix C1.

---

> > ### Comment · Reviewer_67Mn · 2022-08-26
> > **Reponse to response to review**
> >
> > Thank you to the authors for addressing the comments. Overall, my assessment has not changed. The paper reports a systems-level contribution and is well executed but the the degree to which it is sensitive to parameters and heuristics may have some negative result  on how impactful the work is in the community.
> >
> > Thank you!

---

### Author Response · Authors · 2022-08-24
**Response to all reviewers, additional results**

Dear Reviewers,

Thank you for your effort carefully reviewing our paper. We are grateful for your positive comments regarding motivation, novelty, and evaluation, as well as the constructive feedback that helps to improve our paper.

Here we respond to the main suggestions highlighted in the meta review, and provide additional results in the attached pdf. We will also send individual responses to each reviewer to address additional comments.

**1. Validation with a close-loop simulation or real-world experiments**

We fully agree on the value of real-world and/or realistic simulation experiments. However, for the task of driving in dense urban traffic, setting up either real-world or realistic simulation experiments would require substantial effort on their own, and they are not in the intended scope of this work. Real world experiments are hard because of expensive hardware, safety, and legislative requirements. Closed-loop simulation experiments would require realistic simulation of interactive traffic agent behavior, which is not available in popular AV simulators such as CARLA. Developing simulators with realistic agent behavior is an active area of research [52].

We have nevertheless performed a simple additional closed-loop simulation experiment. We used a closed-loop setup where non-ego trajectories are replayed from the dataset (thus non-ego agents do not react to ego actions in a way reminiscent of settings where agents are distracted). Results are in the attached pdf. The results suggest that previously observed benefits of DiffStack in an open-loop setting translate to the closed-loop setting. Despite the limitations of our simulation setup, these results are encouraging.

**2/a. Comparison with a classical modular approach**

We have actually compared with a classical modular approach, results are in the rows “Standard” in Table 1 and Table 3. For a meaningful comparison, our “Standard” stack is built of the classical equivalents of the prediction, planning, and control modules of DiffStack (lines 218-221). That is, in the standard stack there is no backpropagation through modules and only the prediction model is trained.

**2/b Comparison with a non-interpretable end-to-end system**

First, we would like to clarify that settling the debate between classical and end-to-end AV architectures is beyond the scope of this paper. Our claims, and accordingly our experiments, are focused on improving classical modular stacks through joint training. Second, meaningfully comparing classical stacks and end-to-end learned architectures is difficult in general, because the outcome would be highly dependent on the exact architecture (e.g., the sophistication of the modules in the classical stack, the overall complexity of the end-to-end network), the amount of training data, the complexity and diversity of evaluation scenarios, etc. Indeed, many influential prior works on end-to-end approaches lack comparison with modular stacks [13, 14, 15].

Nevertheless, we did perform additional experiments with naive end-to-end policy networks. We used networks that extend our prediction model with a simple additional policy head, and trained them to output both agent predictions and ego controls, similarly in spirit to [13]. The network could learn good predictions but not ego control (hindsight costs were substantially higher than for the standard stack). As discussed above, this does not mean that a more suitable network architecture, potentially using more data, could not learn a better policy. For this reason, we decided not to add results on end-to-end network approaches.

---

### Meta-Review · Area_Chair_BmVH · 2022-08-10

**Recommendation:** Accept (Poster)
**Confidence:** 4

**Metareview:**

This paper proposes a modular but fully differentiable stack for self-driving cars. The differentiability enables the gradient to back-propagate from task performance all the way to the prediction module. It demonstrates better prediction accuracy than the traditional modular approach while preserving interpretability and reusability.

All the reviewers agree that the paper is well motivated, written and executed. The paper tackles an important problem and contains ideas that will potentially have a major impact in the field of autonomous vehicles. Also, there are also a few areas for improvements brought up by the reviewers:
1) Validation with a close-loop simulation or real-world experiments would significantly improve the quality and the potential impact of this paper.
2) Comparison with a classical modular approach and/or a non-interpretable end-to-end system may reveal more benefits or limitations of this work. Such comparisons are worth adding and discussing.

The authors' response and the additional experiments have sufficiently addressed the main concerns in the original reviews. Thus, we would like to recommend accepting this paper.


**Best Paper Nomination:**

No